# Clinically Relevant Dose of Pemafibrate, a Novel Selective Peroxisome Proliferator-Activated Receptor α Modulator (SPPARMα), Lowers Serum Triglyceride Levels by Targeting Hepatic PPARα in Mice

**DOI:** 10.3390/biomedicines10071667

**Published:** 2022-07-11

**Authors:** Zhe Zhang, Pan Diao, Xuguang Zhang, Takero Nakajima, Takefumi Kimura, Naoki Tanaka

**Affiliations:** 1Department of Metabolic Regulation, Shinshu University School of Medicine, Matsumoto 390-8621, Japan; zhangzhe0914@gmail.com (Z.Z.); 20hm126a@shinshu-u.ac.jp (P.D.); 21hm118c@shinshu-u.ac.jp (X.Z.); nakat@shinshu-u.ac.jp (T.N.); 2Department of Gastroenterology, Shinshu University School of Medicine, Matsumoto 390-8621, Japan; kimuratakefumii@yahoo.co.jp; 3Department of Global Medical Research Promotion, Shinshu University Graduate School of Medicine, Matsumoto 390-8621, Japan; 4International Relations Office, Shinshu University School of Medicine, Matsumoto 390-8621, Japan; 5Research Center for Social Systems, Shinshu University, Matsumoto 390-8621, Japan

**Keywords:** β-oxidation, pemafibrate, PPARα, triglycerides, liver, FGF21

## Abstract

Pemafibrate (PEM) is a novel lipid-lowering drug classified as a selective peroxisome proliferator-activated receptor α (PPARα) modulator whose binding efficiency to PPARα is superior to that of fibrates. This agent is also useful for non-alcoholic fatty liver disease and primary biliary cholangitis with dyslipidemia. The dose of PEM used in some previous mouse experiments is often much higher than the clinical dose in humans; however, the precise mechanism of reduced serum triglyceride (TG) for the clinical dose of PEM has not been fully evaluated. To address this issue, PEM at a clinically relevant dose (0.1 mg/kg/day) or relatively high dose (0.3 mg/kg/day) was administered to male C57BL/6J mice for 14 days. Clinical dose PEM sufficiently lowered circulating TG levels without apparent hepatotoxicity in mice, likely due to hepatic PPARα stimulation and the enhancement of fatty acid uptake and β-oxidation. Interestingly, PPARα was activated only in the liver by PEM and not in other tissues. The clinical dose of PEM also increased serum/hepatic fibroblast growth factor 21 (FGF21) without enhancing hepatic lipid peroxide 4-hydroxynonenal or inflammatory signaling. In conclusion, a clinically relevant dose of PEM in mice efficiently and safely reduced serum TG and increased FGF21 targeting hepatic PPARα. These findings may help explain the multiple beneficial effects of PEM observed in the clinical setting.

## 1. Introduction

Triglyceride (TG) is a prominent energy source, having the largest reserves and production capacity in the body. TG becomes hydrolyzed into glycerol and free fatty acid (FFA). FFA is also absorbed from the intestine into the circulation for use in peripheral tissues, with any surplus stored as TG in white adipose tissue (WAT) [1,2]. Overnutrition and ensuing obesity, dyslipidemia, insulin resistance, diabetes, and non-alcoholic fatty liver disease (NAFLD) can cause significant glucolipotoxicity and other disorders harmful to human health, including atherosclerosis and cancer, and are associated with higher overall mortality. Hypertriglyceridemia-related diseases are now widespread and have become a global issue requiring immediate and appropriate intervention [3,4,5,6,7].

Currently available TG-lowering drugs include bile acid-binding resins, n-3 polyunsaturated fatty acid (FA), nicotinic acid, and fibrates. Fibrates are agonists of peroxisome proliferator-activated receptor α (PPARα) [6,8,9,10]. PPARα is the key transcription factor controlling FA transport and β-oxidation in peroxisomes and mitochondria, and attenuate inflammatory signaling and cell stress [6,11]. However, fibrates might also modulate nuclear receptors other than PPARα, such as pregnane X receptor (PXR) [12]. Additionally, our earlier studies revealed that clinically relevant doses of bezafibrate (10 mg/kg/day), fenofibrate (5 mg/kg/day), and clofibrate (15 mg/kg/day) could not sufficiently activate PPARα in mice [13]. Considering the beneficial effects of PPARα, more efficient and selective PPARα activators are desired.

Kowa Company, Ltd., has recently developed a new PPARα agonist designated as a selective PPARα modulator (SPPARMα). This drug, pemafibrate (PEM, K-877, Parmodia^™^), is currently approved in Japan for the treatment of hyperlipidemia. PEM has higher selectivity and approximately 2500-fold potency for activating human PPARα versus the conventional fibrate fenofibrate. Although hepatotoxicity, renotoxicity, and rhabdomyolysis are the main adverse effects of conventional fibrates, the frequency of these events is reportedly low for PEM [11,14,15,16,17]. The usefulness of PEM for dyslipidemic patients with NAFLD or primary biliary cholangitis have documented [18]. Moreover, PEM can be used in patients with impaired renal function due to its excretion in the bile [19,20,21].

The recommended clinical dose of PEM is 0.1 mg twice a day, with a maximum dose of 0.4 mg/day [9,14]. The clinical dose of PEM was determined based on repeated administrations to mice demonstrating a similar maximum concentration and area under the concentration-time curve of PEM to those in humans. When 0.2 mg of PEM was administered twice daily to healthy adults for 1 week, the maximum plasma concentration (Cmax) and area under the concentration-time curve (AUC) of PEM were 3.572 ± 1.021 ng/mL and 12.207 ± 2.900 ng·h/mL, respectively. In mice, the administration of 0.075 mg/kg and 0.1 mg/kg of PEM for 4 weeks yielded respective mean Cmax values of 2.94 and 3.0 ng/mL and mean AUC values of 11.1 and 14.5 ng·h/mL. These pharmacodynamic indicators revealed the PEM doses in mice providing comparable Cmax and AUC values in clinical practice to be 0.075–0.1 mg/kg/day. However, the dose of PEM in previous mouse experiments was often more than 10-fold (i.e., 1–10 mg/kg/day); thus, the exact mechanism of how PEM reduces serum TG in the clinical setting has not been fully evaluated in mice [22,23,24].

To address this issue, this present study administered clinically relevant (0.1 mg/kg/day) or relatively high (0.3 mg/kg/day) doses of PEM to mice for evaluations of drug efficacy and safety as well as to clarify the real-world mechanism of the TG-lowering effect of PEM.

## 2. Materials and Methods

### 2.1. Mice and Treatment

Male 8-week-old mice on a C57BL/6J genetic background weighing 20–25 g were purchased from CLEA Japan, Inc. (Tokyo, Japan) and maintained under controlled conditions (25 °C, 12 h light/dark cycle, 4–6 mice/cage) with tap water and standard laboratory chow ad libitum [13]. The mice were randomly divided into three groups: the control group (*n* = 8), the clinically relevant dose (0.1 mg/kg/day) PEM group (PEM-0.1 group; *n* = 10), and the relatively high dose (0.3 mg/kg/day) PEM group (PEM-0.3 group; *n* = 9). The mice were fed a normal diet during the acclimatization period and weighed daily before treatment. PEM (Kowa Company, Ltd., Tokyo, Japan) was suspended in 0.5% (*w*/*v*) carboxymethylcellulose (Wako Pure Chemical Industries, Ltd., Osaka, Japan) at final concentrations of 0.01 and 0.03 mg/mL for the 0.1 and 0.3 mg/kg/day treatments, respectively, and suspensions were administered by oral gavage once daily at approximately 10:00 am for 2 weeks. The same amount of 0.5% (*w*/*v*) carboxymethylcellulose without PEM was administered to control mice as a vehicle in a similar manner. On the 14th day after the last gavage at 9:30 am, the mice were fasted for 4 h and sacrificed under anesthesia for the collection of blood and tissues. The plasma samples were centrifuged at low speed (3000× *g*) for 20 min, and the supernatant was collected for recentrifugation at high speed (10,000× *g*) for 5 min, after which the supernatant was taken to obtain test serum. After dissection, the harvested organ tissues along with the serum samples were cryogenically stored at −80 °C for future assays. All animal experiments were conducted in accordance with the National Academy of Sciences. The animal study protocol (#290026 (1 August 2017), #300037 (12 July 2018)) was approved by the Shinshu University School of Medicine “Guide to the Care and Use of Experimental Animals”.

### 2.2. Histological Analysis

Formalin-fixed tissues of the liver, kidney, heart, brown adipose tissue (BAT), and epididymal WAT (eWAT) were embedded in paraffin, sliced into 4 μm sections, and stained with hematoxylin and eosin for histopathological examination under light microscopy. A minimum of three discrete sections were evaluated per mouse for each tissue [25].

### 2.3. Biochemical Analysis

Serum alanine aminotransferase (ALT), aspartate aminotransferase (AST), TG, total cholesterol (T-Chol), non-esterified fatty acid (NEFA), phospholipid (PL), and glucose were measured with commercial assay kits (Wako Pure Chemical Industries, Ltd.). Serum insulin concentrations were determined by means of a mouse insulin enzyme-linked immunosorbent assay (ELISA) kit (AKRIN-011T, company name, Gunma, Japan). Serum high molecular weight (HMW) adiponectin was quantified using a mouse/rat adiponectin ELISA kit (AKMAN-011, Gunma, Japan). Serum and liver fibroblast growth factor 21 (FGF21) levels were assayed by a mouse FGF21 ELISA kit (RSD MF2100, R&D Systems, Minneapolis, MN, USA). To extract total liver lipids, approximately 50 mg of frozen liver tissue was cut and sonicated in 5–10 volumes of sodium phosphate buffer (NaPi, 50 mM) [26]. The lysate (50 μL) was transferred to a long tube for the addition of 900 μL of hexane/isopropanol (3:2) (HIP). The mixture was vortexed vigorously for 1 min and centrifuged at 4 °C at 2500 rpm for 5 min. The upper layer was transferred to a 1.5 mL tube and centrifuged under vacuum at 40–50 °C. After drying, the total liver lipid precipitate formed a gel-like substance or white particles on the sides and bottom of the tube. Then, 100 μL of HIP with Triton X-100 was added to the sample, which was dissolved and evaporated again in a vacuum centrifuge. Eventually, 100 μL of distilled water was added to the sample prior to incubation at 37 °C for 30 min [27]. The total liver lipids were assayed with a commercial assay kit (Wako Pure Chemical Industries, Ltd.).

### 2.4. Analysis of mRNA Expression

Approximately 25 mg of liver or kidney tissue were homogenized, and total RNA was extracted with NucleoSpin RNA Plus (MACHEREY-NAGEL GmbH & Co. KG, Düren, Germany). The same amounts of heart tissue, BAT, and eWAT were homogenized in 0.5 mL of TRI Reagent (MOR Molecular Research Center, Inc., Cincinnati, OH, USA) using a glass homogenizer, and the colorless upper layer was transferred to a 1.5 mL tube followed by the addition of the same volume of 75% ethanol. After vortexing, the mixture was transferred to a NucleoSpin^®^ RNA Plus Column and subjected to RNA extraction. RNA quality was assessed by Nanodrop 2000 measurement (Thermo Fisher Scientific Inc., Waltham, MA, USA), and the RNA was reverse transcribed into cDNA with ReverTra Ace^®^ qPCR RT Master Mix (Toyobo Co., Ltd., Osaka, Japan). Next, mRNA levels were determined by the real-time quantitative polymerase chain reaction (qPCR) using a THUNDERBIRD SYBR qPCR Mix (Toyobo Co., Ltd.) on a Thermo Fisher QuantStudio 3 Real-Time PCR System (Thermo Fisher Scientific Inc.). The primer sequences used for qPCR were designed according to the BLAST database released from the US National Library of Medicine (Appendix A) [2,28]. qPCR was carried out in 96-well plates with 1 μL of each cDNA sample. mRNA levels were quantified using the comparative Ct method, normalized to that of 18S ribosomal RNA (18S rRNA), and then expressed as fold changes relative to those of vehicle-treated control C57BL/6 mice.

### 2.5. Immunoblot Analysis

Approximately 25 mg of liver tissue were homogenized with protein lysis buffer (sucrose 0.25 M, Tris chloride 25 mM, KCl 25 mM, MgCl_2_ 5 mM, 0.5% (*w*/*v*) Triton X-100, DTT 1 mM, pH 7.4) containing protease and phosphatase inhibitor (100-fold dilution, Thermo Fisher Scientific Inc.). Approximately 50 mg of liver tissue were transferred to a chilled Dounce homogenizer (Wheaton, Milliville, NJ, USA) and homogenized on ice by 20 strokes. Liver nuclear fractions were isolated using NE-PER^®^ Nuclear and Cytoplasmic Extraction Reagents (Thermo Fisher Scientific Inc.) containing protease and phosphatase inhibitor (100-fold dilution, Thermo Fisher Scientific Inc.). Protein concentrations were measured colorimetrically with a BCA Protein Assay kit (Pierce, Rockford, IL, USA) [29]. Whole liver lysates (40–60 μg protein in each lane) were separated by 7.5–15% sodium dodecyl sulfate-polyacrylamide gel electrophoresis (SDS-PAGE), which was dependent on the molecular weight of the target protein for western blot analysis. Nuclear fractions (40 μg protein in each lane) were similarly separated using 12.5% SDS-PAGE. After electrophoresis, the proteins were transferred to polyvinylidene fluoride membranes (IPVH00010, Merck Millipore Ltd., Munich, Germany) or nitrocellulose filter membranes (10600016, Merck Millipore Ltd.). At room temperature, the membranes were blocked with 3–10% skimmed milk powder, in Tris buffer for 1 h, and then incubated overnight at 4 °C with the respective primary antibodies listed in Appendix A [30]. After washing with Tris-buffered saline containing Tween 20, the membranes were incubated with alkaline phosphatase-conjugated secondary antibodies (4000-fold dilution, No. 93785, Jackson ImmunoResearch Laboratories, West Grove, PA, USA) with 1-step NBT/BCIP substrate (Pierce, Rockford, IL, USA) or horseradish peroxidase-conjugated secondary antibodies (4000-fold dilution, No. 115-035-003, Jackson ImmunoResearch Laboratories). The position of the protein band was determined by its molecular weight. Finally, the Bio-Rad ChemiDoc Touch (Bio-Rad Laboratories, Inc., Hercules, CA, USA) was used to detect the strength of the chemiluminescence signal of the target protein. The actual position of the protein band was determined using co-electrophoresis molecular weight standards (PM2500, Smobio, Hsinchu, Taiwan) together with the molecular weight [13,25,29]. Each target protein was subjected to a minimum of two immunoblotting analyses to ensure the same trend of change. The intensity of each band was quantified using NIH Image J software (National Institutes of Health, Bethesda, MD, USA), normalized to that of a loading control, and expressed as a fold change relative to that of the vehicle-treated control group. Band intensity was quantified densitometrically after normalization to that of β-actin (ACTB) or histone H1 as a loading control.

### 2.6. Statistical Analysis

The results are expressed as the mean ± standard deviation (SD). The two-tailed Student’s *t*-test was employed for comparisons between the control and the clinically relevant dose or relatively high dose PEM using SPSS statistical software version 22 (IBM, Armonk, NY, USA). A *p*-value of less than 0.05 was considered statistically significant.

## 3. Results

### 3.1. Lack of Increased Serum AST and ALT at Clinically Relevant Dose of PEM Treatment in Mice

All test mice appeared healthy, with no change in food intake during the 2 weeks of PEM treatment. At the end of the study, the mice in the PEM-0.1 group had gained weight. Liver/body weight was also significantly increased in PEM group mice versus control mice. The liver specimens from the PEM-0.3 group tended to be larger, darker, and heavier than those from the control and PEM-0.1 groups. The amount of eWAT was significantly decreased in the PEM-0.3 group (Figure 1A,B). Histological liver analysis revealed no differences between the PEM-0.1 group and control mice, whereas binuclear hepatocytes, narrowing of liver sinusoidal spaces, and bizarre eosinophilic hepatocytes were detected in samples from the PEM-0.3 group (Figure 1C,D). Marked increases in mitotic figures were not seen in the livers of PEM-0.3 mice. Indeed, serum AST and ALT levels as indicators of liver injury were significantly increased in the high-dose PEM-treated animals (Figure 2A). The clinically relevant dose of PEM, therefore, appeared to be more tolerated and suitable for mouse experiments, owing to the absence of apparent liver injury.

### 3.2. Clinically Relevant Dose of PEM Reduces Serum TG and NEFA in Mice

Serum lipids and glucose were determined next. Serum T-Chol was significantly increased in PEM-0.1 mice as compared with the control group, while serum TG and NEFA were significantly decreased in both PEM-treated groups versus controls (Figure 2A). Serum PL, glucose, and adiponectin were similar among the groups, although serum insulin levels were significantly reduced in the PEM-0.3 group (Figure 2A, Appendix A). Hepatic T-Chol, TG, and NEFA were unchanged by either PEM treatment (Figure 2B). These results indicated that the clinically relevant dose of PEM effectively and sufficiently reduced serum TG and NEFA without hepatotoxicity in a manner comparable to when administered to humans.

### 3.3. Clinically Relevant Dose of PEM Increases FA Utilization in the Liver

The liver is a main organ of FA/TG metabolism. To understand the precise mechanisms of lowered serum TG and NEFA by a clinically relevant dose of PEM, we analyzed the hepatic expression of genes related to FA/TG metabolism. The mRNA expression of genes encoding FA translocase (*Cd36*) and FA transport protein 1 (*Slc27a1*), both of which playing a vital role in FA uptake from the blood to the liver, were significantly increased in both PEM-treated groups. A similar increase was observed in the expression of mRNA encoding liver-type FA-binding protein (L-FABP, *Fabp1*) associated with intracellular FA transport (Figure 3A). Immunoblot analysis demonstrated that PEM increased L-FABP and long-chain acyl-coenzyme A synthase (LACS), a contributor to FA activation and subsequent FA-CoA generation (Figure 3B). Based on these findings, PEM appeared to accelerate FA utilization in the liver.

### 3.4. Clinically Relevant Dose of PEM Enhances Mitochondrial and Peroxisomal FA β-Oxidation

Since FA is metabolized through mitochondrial and peroxisomal β-oxidation, the expression levels of genes related to FA β-oxidation were determined. In the pathway for FA-CoA entrance into mitochondria, the mRNA expression of mitochondrial carnitine/acylcarnitine carrier protein (*Slc25a20*) was significantly increased in both PEM groups. Indeed, a clinically relevant dose of PEM drastically increased mRNA levels encoding mitochondrial FA β-oxidizing enzymes, including short-chain acyl-CoA dehydrogenase (SCAD, *Acads*), medium-chain acyl-CoA dehydrogenase (MCAD, *Acadm*), long-chain acyl-CoA dehydrogenase (LCAD, *Acadl*), and very-long-chain acyl-CoA dehydrogenase (VLCAD, *Acadvl*) (Figure 4A). These changes in FA β-oxidation enzymes were confirmed by immunoblot analysis (Figure 4B).

The mRNA expression of genes encoding acyl-CoA oxidase 1 (ACOX1, *Acox1*), peroxisomal bifunctional protein (PH, *Ehhadh*), and peroxisomal thiolase (PT, *Acaa1*), all of which are prominent peroxisomal FA β-oxidation enzymes, was markedly increased in PEM-treated mice (Figure 5A). These changes were confirmed by immunoblot analysis, with the induction of peroxisomal β-oxidation enzymes comparable between the PEM-0.1 and PEM-0.3 groups (Figure 5B). Thus, a clinically relevant dose of PEM could effectively and sufficiently activate FA β-oxidation in mitochondria and peroxisomes, thereby leading to reduced serum TG.

### 3.5. Effect of PEM on FA/TG Synthesis and VLDL Secretion

The expression of genes related to FA/TG synthesis and secretion were also quantified. We observed no significant differences in the mRNA levels of genes related to de novo FA synthesis (FA synthase [FAS, *Fasn*] and acetyl-CoA carboxylase α [ACCα, *Acaca*]) (Figure 6A), which was in agreement with immunoblot results (Figure 6B). The mRNA levels of genes involved in TG synthesis (diacylglycerol acyltransferase [DGAT] 1 and 2) and TG hydrolysis (patatin-like phospholipase domain containing 2 [*Pnpla2*] and hepatic lipase [*Lipc*]) were similar between PEM-0.1 and control mice. Although levels of mRNA related to very-low-density lipoprotein (VLDL) secretion, including apolipoprotein B (*Apob*) and microsomal TG transfer protein (MTP, *Mttp*), were unchanged by PEM administration (Figure 6C), marked up-regulation of MTP, a rate-limiting protein to release VLDL particles from hepatocytes into the circulation, was detected by immunoblot analysis (Figure 6D). PEM, therefore, presumably promoted the secretion of VLDL particles from the liver.

### 3.6. Clinically Relevant Dose of PEM Does Not Affect FA/TG Metabolism in Adipose Tissue

FA/TG metabolism in eWAT can influence serum TG/NEFA levels. No significant differences were detectable in the histological features of white adipocytes between PEM-0.1 and control mice (Appendix A). The mRNA expression of genes involved in FA uptake (*Cd36*), FA synthesis (*Fasn* and *Acaca*), TG synthesis (*Dgat1* and *Dgat2*), lipolysis (lipoprotein lipase [*Lpl*], adipose triglyceride lipase [*Atgl*], hormone-sensitive lipase [*Hsl*], carboxylesterase 3 [*Ces3*], and *Pnpla2*), FA β-oxidation (*Acadm*, *Acadl*, and *Acox1*), and browning (peroxisome proliferator-activated receptor gamma coactivator 1 alpha [*Ppargc1a*] and uncoupling protein 1 [*Ucp1*]) were all unaltered by PEM treatment (Figure 7 and Appendix A).

Similarly, there were no significant differences in the histological findings or mRNA levels of genes involved in FA/TG metabolism by PEM treatment (Figure 7 and Appendix A). Taken together, the contribution of eWAT and BAT to reduced serum FA/TG appeared to be minor.

### 3.7. Clinically Relevant Dose of PEM Activates Hepatic PPARα Only and Increases FGF21 without Enhancing Oxidative Stress

FA β-oxidation is regulated by PPARα, a nuclear receptor abundantly expressed in the liver, BAT, heart, and kidney. *Acadm*, *Acadl*, and *Acox1* are typical target genes for PPARα possessing as PPARα response elements. The lack of induction of *Acadm*, *Acadl*, and *Acox1* mRNA levels in BAT prompted us to consider that PPARα activation by PEM occurred only in specific organs. Indeed, no induction of PPARα target genes was detected in PPARα-rich tissues other than in the liver (Figure 4, Figure 5 and Figure 7 and Appendix A). Therefore, the clinical PEM dose exerted its lipid-lowering effect almost exclusively through the activation of hepatic PPARα.

PEM increased hepatic nuclear PPARα content as well as 70 kDa peroxisomal membrane protein (PMP70), an indicator of peroxisomal proliferation (Figure 8A). In contrast, the agent did not increase the expression of PPARβ or PPARγ in the hepatic nuclear fraction (Appendix A). The expression of catalase, an antioxidant enzyme rich in peroxisomes, was also increased by PEM administration (Figure 8B). When PPARα is intensely activated, there is concern of a risk of increased oxidative stress and ensuing cytotoxicity. However, hepatic 4-hydroxy-nonenal (4-HNE)-modified protein, a major lipid peroxidation aldehyde, did not increase by PEM treatment (Figure 8C). In accordance with PPARα activation, the level of FGF21, a hepatokine with protective roles in several tissues that improves lipid/glucose metabolism, was markedly elevated in the liver and serum by PEM (Figure 8D). Our cumulative data showed that a clinical dose of PEM could reduce serum TG/FA and increase FGF21 in mice by hepatic PPARα activation, without inducing liver injury or enhancing oxidative stress.

## 4. Discussion

The results of the present study clearly demonstrated that a clinical dose of PEM could activate PPARα in the liver, enhance FA utilization and β-oxidation, and consequently, reduce serum TG without any apparent adverse effects. No other activation of PPARα was detected in the heart, kidney, or BAT. PEM also increased FGF21 level in the serum and liver. Our findings provide key insights into the mechanism by which PEM safely reduces TG without liver dysfunction in the clinical context (Figure 9).

This study confirmed the strong potential of PEM for activating hepatic PPARα and enhancing β-oxidation at a clinical dose. Although it was reasonable that hepatic PPARα activation led to reduced FA/TG in serum, an unexpected finding was a concomitant increase in MTP expression. MTP is a multifunctional protein involved in the transfer of neutral lipids. MTP also maintains the physiological and regulatory effects of lipid and lipoprotein homeostasis. The drop in serum TG along with an MTP increase indicates accelerated turnover of VLDL [31]. Since decreased MTP expression has been reported in NAFLD, especially in non-alcoholic steatohepatitis [32,33,34], the combination of increased MTP and augmented β-oxidation might attenuate hepatic fat accumulation.

The protective effects of PEM have been documented in extrahepatic organs both in mouse experiments and in the clinical setting. For example, PEM promoted ischemia-induced revascularization through eNOS-dependent mechanisms in mice [35,36]. PEM also attenuated neointima formation after vascular injury in mice fed normal chow and a high-fat diet [36,37,38]. However, according to the results of the current study, those favorable PEM effects may not have been caused by the direct activation of PPARα in extrahepatic tissues, such as the endothelium and vascular tissue. One possible explanation of how PEM exerts systemic protective effects is a reduction in atherogenic lipids, which disrupt endothelial function and microcirculation [4,24,39,40,41]. Another possibility is mediation of the beneficial effects of PEM throughout the body by FGF21. FGF21 can lower blood cholesterol, TG, and low-density lipoprotein, while increasing blood high-density lipoprotein [28,42,43,44,45,46]. In the cardiovascular system, FGF21 exerts a protective function on endothelial cells by resisting oxidative stress, inhibiting aberrant vascular remodeling, and antagonizing angiotensin II [47,48,49,50]. Studies have demonstrated that FGF21 can also significantly reduce fasting blood glucose level and improve glucose clearance without hypoglycemia [51,52,53,54]. Significant FGF21 production in the liver and enhanced circulating FGF21 may, therefore, contribute to the beneficial effects of PEM.

The recent studies demonstrated that PEM reduced serum TG and induced hepatic β-oxidation-related genes in mice treated with high-fat diet [55,56]. Although we used a normal mice and diet in this study, the results were basically consistent with the findings obtained from high-fat diet-fed mice. Therefore, we assume that PEM exerts hypolipidemic effect in patients with hyperlipidemia in a similar mechanism.

Persistent and potent PPARα activation leads to hepatomegaly and liver injury in mice [57,58,59]. Foreman et al. demonstrated that potent, high-affinity human PPARα agonist GW7647 induced hepatomegaly and increased Myc expression and hepatic cytotoxicity and necrosis in wild-type mice [60]. These effects did not occur or were largely diminished in *Ppara*-disrupted and *PPARA*-humanized mice, indicating the existence of species difference in the hepatotoxic effects of PPARα activators. We found that PEM at the relatively high dose (0.3 mg/kg/day) caused serum ALT increase and histological alterations, such as sinusoidal narrowing, increased binuclear hepatocytes, and bizarre eosinophilic hepatocytes. There were no obvious data regarding hepatotoxicity in the previous studies of mice treated with more than 0.3 mg/kg/day of PEM [22,23,24,38,61], while apparent hepatotoxicity was not reported in the previous mouse studies at 0.1 mg/kg/day or less of PEM [55,56], which is consistent with the results in the current study. Therefore, the clinically relevant dose of PEM (0.1 mg/kg/day) used in the present study is likely appropriate to evaluate the action of PEM using mice, from the viewpoint of not only pharmacodynamic similarity to humans but also hepatotoxicity.

Although this investigation was designed to reproduce the circumstances of PEM-treated hyperlipidemic patients, it had several limitations. First, the PEM treatment period was relatively short. The efficacy and safety of long-term, clinically relevant PEM dosage is needed in various animal disease models. Second, the mode of PPARα activation differs slightly between humans and mice; further experiments using animals humanized for PPARα genes or human samples are required. Third, PEM treatment with liver-specific *Ppara* mice will solve the question of whether reduced circulating TG by PEM is truly mediated by hepatic PPARα only. Lastly, we set the clinically relevant dose of PEM based on the pharmacodynamic parameters, such as AUC and Cmax. However, the Reagan-Shaw formulae or similar equations which adjust for body weight or body surface area is also available for conversion of human doses to experimental animals [62]. Dose setting of PEM based on body surface area deserves further investigation.

## 5. Conclusions

A clinically relevant dose of PEM could efficiently and safely reduce serum TG and increase FGF21 targeting of hepatic PPARα in mice. These findings may help explain the precise mechanism of how PEM attenuates circulating TG in humans.

## Figures and Tables

**Figure 1 biomedicines-10-01667-f001:**
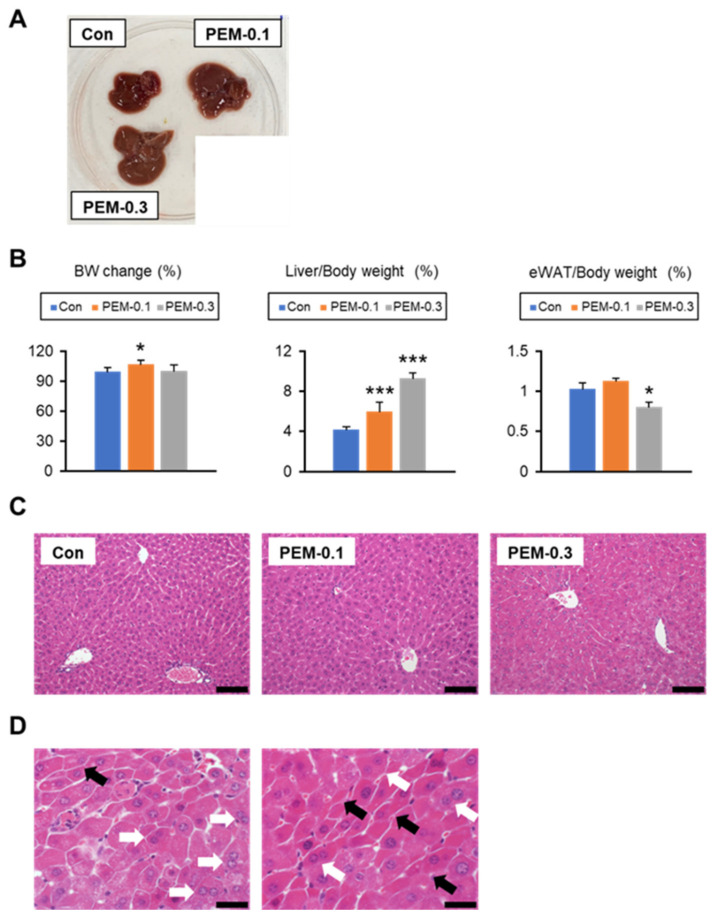
Phenotypic changes after 2-week PEM treatment at a clinically relevant dose (0.1 mg/kg/day) or relatively high dose (0.3 mg/kg/day). (**A**) Gross appearance of the liver. (**B**) Body weight (BW) change and the ratio of liver and epididymal white adipose tissue (eWAT) weight to BW. (**C**) Representative photomicrographs of hematoxylin and eosin-stained liver sections. Scale bar = 50 μm. (**D**) Hepatic sinusoidal narrowing (black arrow) and binuclear cells (white arrow) were found in liver tissue sections in the relatively high PEM group. Scale bar = 20 μm. Data are expressed as the mean ± SD. * *p* < 0.05 and *** *p* < 0.001 vs. control group. Con, vehicle-treated mice; PEM-0.1, pemafibrate-treated mice at a clinically relevant dose (0.1 mg/kg/day); PEM-0.3, pemafibrate-treated mice at a relatively high dose (0.3 mg/kg/day).

**Figure 2 biomedicines-10-01667-f002:**
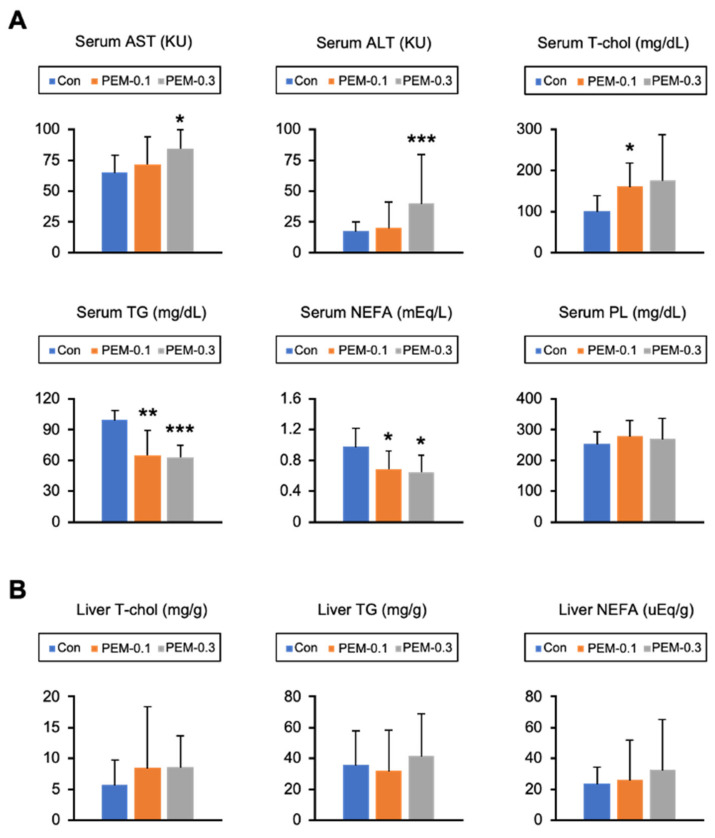
Clinically relevant dose of PEM reduces serum TG and NEFA. (**A**) Serum aspartate aminotransferase (AST) and alanine aminotransferase (ALT) activities and serum total cholesterol (T-Chol), triglyceride (TG), non-esterified fatty acid (NEFA), and phospholipid (PL) levels. (**B**) Hepatic content of T-Chol, TG, and NEFA. Data are expressed as the mean ± SD. * *p* < 0.05, ** *p* < 0.01, and *** *p* < 0.001 vs. control group. Con, vehicle-treated mice; PEM-0.1, pemafibrate-treated mice at a clinically relevant dose (0.1 mg/kg/day); PEM-0.3, pemafibrate-treated mice at a relatively high dose (0.3 mg/kg/day).

**Figure 3 biomedicines-10-01667-f003:**
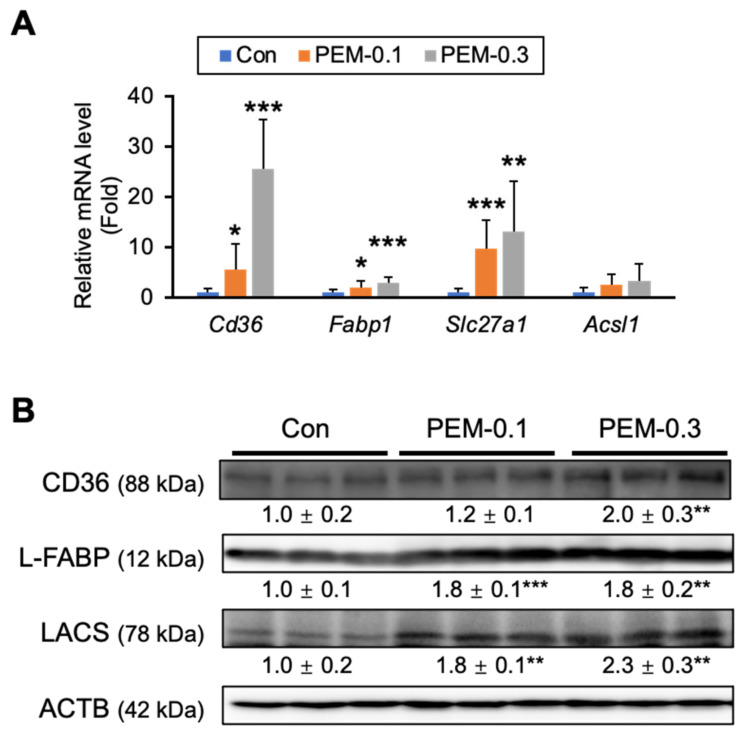
Clinically relevant dose of PEM significantly increases FA uptake in the liver. (**A**) Hepatic mRNA levels of *Cd36*, *Fabp1*, *Slc27a1*, and *Acsl1* were quantified by qPCR, normalized to that of 18s ribosomal RNA, and expressed as values relative to those of control mice. (**B**) Immunoblot analysis of CD36, L-FABP, and LACS. ACTB was used as a loading control. Band intensity was measured densitometrically, normalized to that of ACTB, and expressed as values relative to those of control mice. Results were obtained from two independent immunoblot experiments. Data are expressed as the mean ± SD. * *p* < 0.05, ** *p* < 0.01, and *** *p* < 0.001 vs. control group. Con, vehicle-treated mice; PEM-0.1, pemafibrate-treated mice at a clinically relevant dose (0.1 mg/kg/day); PEM-0.3, pemafibrate-treated mice at a relatively high dose (0.3 mg/kg/day).

**Figure 4 biomedicines-10-01667-f004:**
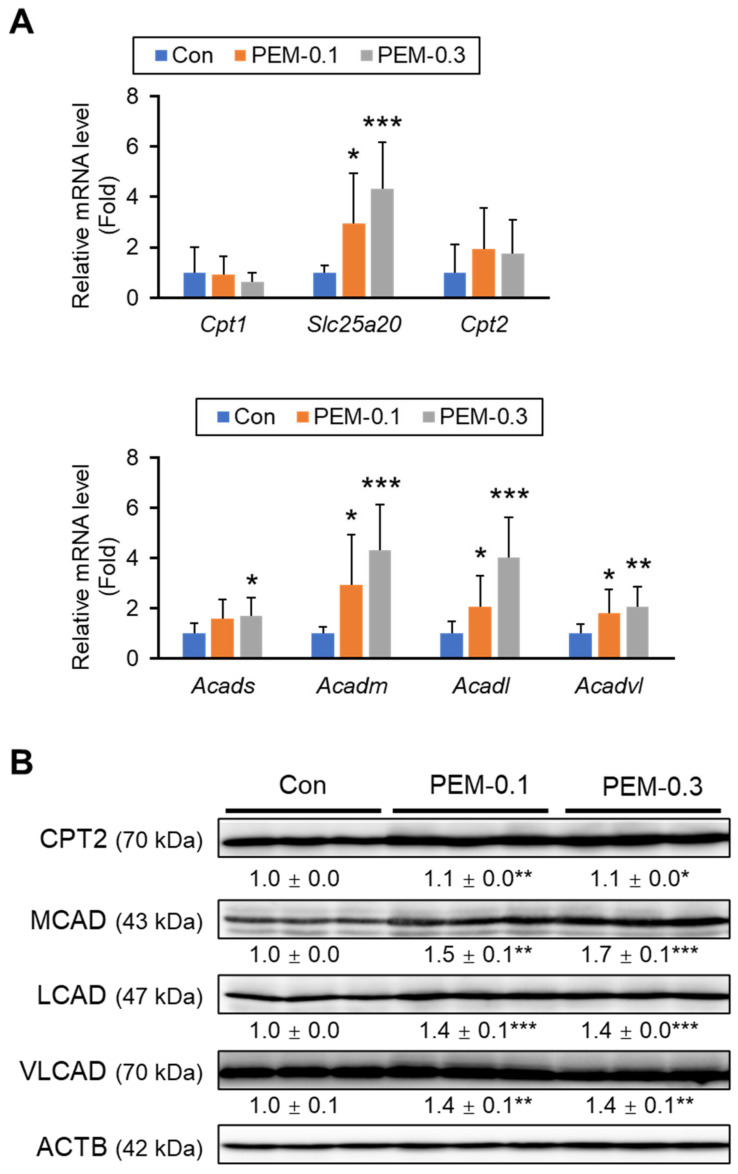
PEM augments mitochondrial FA β-oxidation in the liver. (**A**) Hepatic mRNA levels of genes related to mitochondria FA β-oxidation (*Cpt1*, *Slc25a20*, *Cpt2*, *Acads*, *Acadm*, *Acadl*, and *Acadvl*) were quantified by qPCR, normalized to that of 18s ribosomal RNA, and expressed as values relative to those of control mice. (**B**) Immunoblot analysis of CPT2, MCAD, LCAD, and VLCAD. ACTB was used as a loading control. Band intensity was measured densitometrically, normalized to that of ACTB, and expressed as values relative to those of control mice. Results were obtained from two independent immunoblot experiments. Data are expressed as the mean ± SD. * *p* < 0.05, ** *p* < 0.01, and *** *p* < 0.001 vs. control group. Con, vehicle-treated mice; PEM-0.1, pemafibrate-treated mice at a clinically relevant dose (0.1 mg/kg/day); PEM-0.3, pemafibrate-treated mice at a relatively high dose (0.3 mg/kg/day).

**Figure 5 biomedicines-10-01667-f005:**
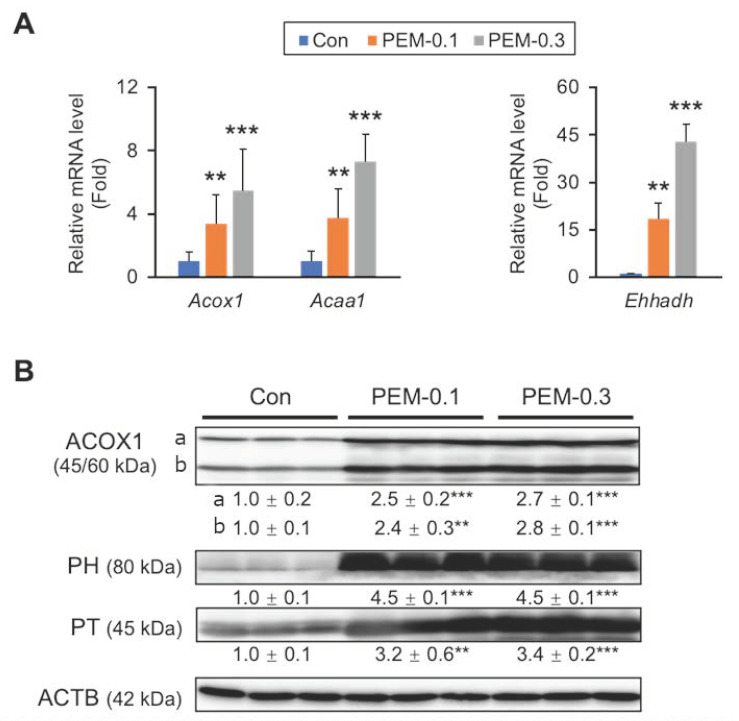
PEM augments peroxisomal FA β-oxidation. (**A**) Hepatic mRNA levels of genes related to peroxisome FA β-oxidation (*Acox1*, *Acaa1*, and *Ehhadh*) were quantified by qPCR, normalized to that of 18s ribosomal RNA, and expressed as values relative to those of control mice. (**B**) Immunoblot analysis of ACOX1, PH, and PT. ACTB was used as a loading control. Band intensity was measured densitometrically, normalized to that of ACTB, and expressed as values relative to those of control mice. Results were obtained from two independent immunoblot experiments. Data are expressed as the mean ± SD. ** *p* < 0.01 and *** *p* < 0.001 vs. control group. Con, vehicle-treated mice; PEM-0.1, pemafibrate-treated mice at a clinically relevant dose (0.1 mg/kg/day); PEM-0.3, pemafibrate-treated mice at a relatively high dose (0.3 mg/kg/day).

**Figure 6 biomedicines-10-01667-f006:**
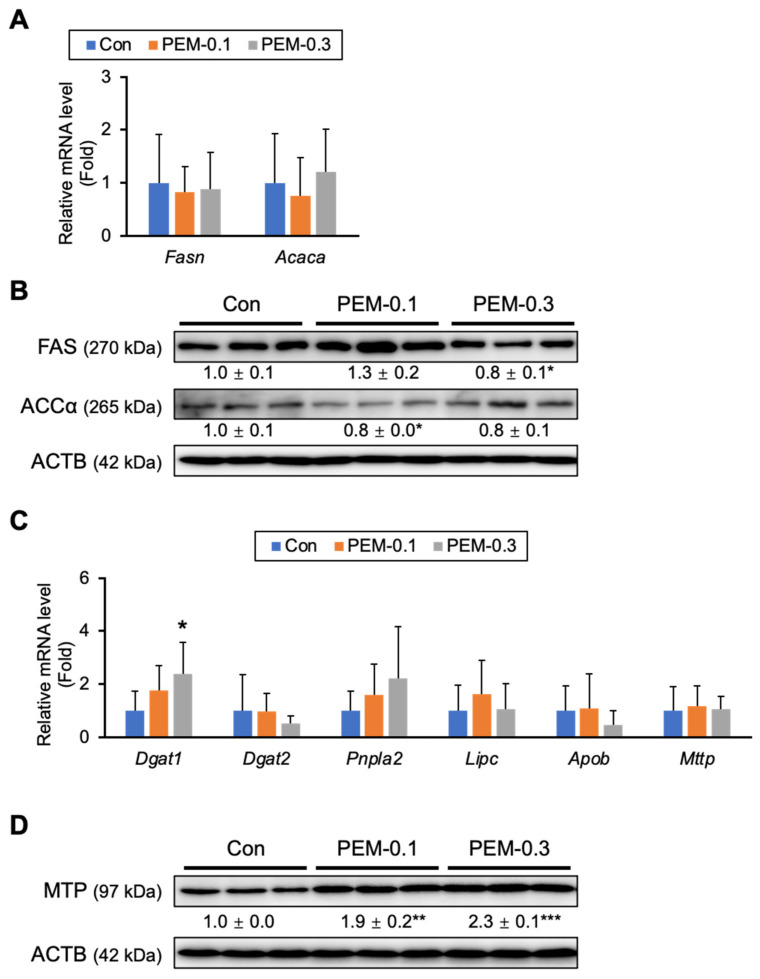
Effect of PEM on de novo FA synthesis, TG synthesis, and VLDL secretion. (**A**,**C**) Hepatic mRNA levels of genes related to FA de novo synthesis (*Fasn* and *Acaca*), TG synthesis (*Dgat1* and *Dgat2*), FA hydrolysis (*Pnpla2* and *Lipc*), and FA secretion into the circulation (*Apob* and *Mttp*) were quantified by qPCR, normalized to that of 18s ribosomal RNA, and expressed as values relative to those of control mice. (**B**,**D**) Immunoblot analysis of FAS, ACCα, and MTP. ACTB was used as a loading control. Band intensity was measured densitometrically, normalized to that of ACTB, and expressed as values relative to those of control mice. Results were obtained from two independent immunoblot experiments. Data are expressed as the mean ± SD. * *p* < 0.05, ** *p* < 0.01, and *** *p* < 0.001 vs. control group. Con, vehicle-treated mice; PEM-0.1, pemafibrate-treated mice at a clinically relevant dose (0.1 mg/kg/day); PEM-0.3, pemafibrate-treated mice at a relatively high dose (0.3 mg/kg/day).

**Figure 7 biomedicines-10-01667-f007:**
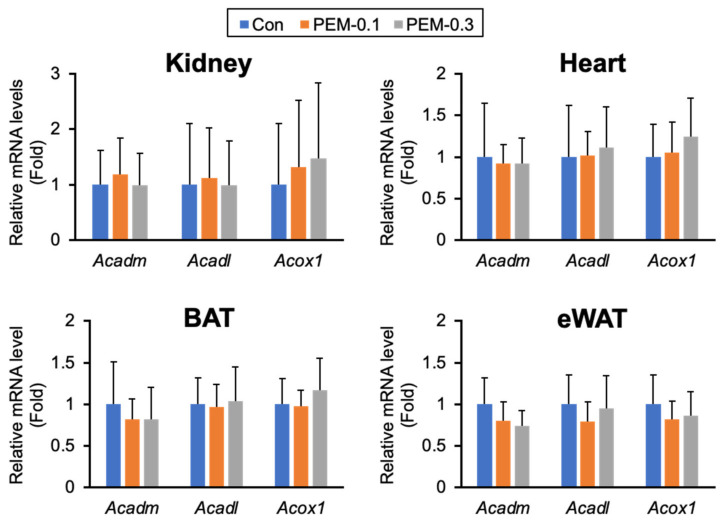
Clinically relevant dose of PEM does not induce PPARα target genes in extrahepatic tissues. Expression levels of PPARα target genes (*Acadm*, *Acadl*, and *Acox1*) in the kidney, heart, BAT, and eWAT. Data are expressed as the mean ± SD. Con, vehicle-treated mice; PEM-0.1, pemafibrate-treated mice at a clinically relevant dose (0.1 mg/kg/day); PEM-0.3, pemafibrate-treated mice at a relatively high dose (0.3 mg/kg/day).

**Figure 8 biomedicines-10-01667-f008:**
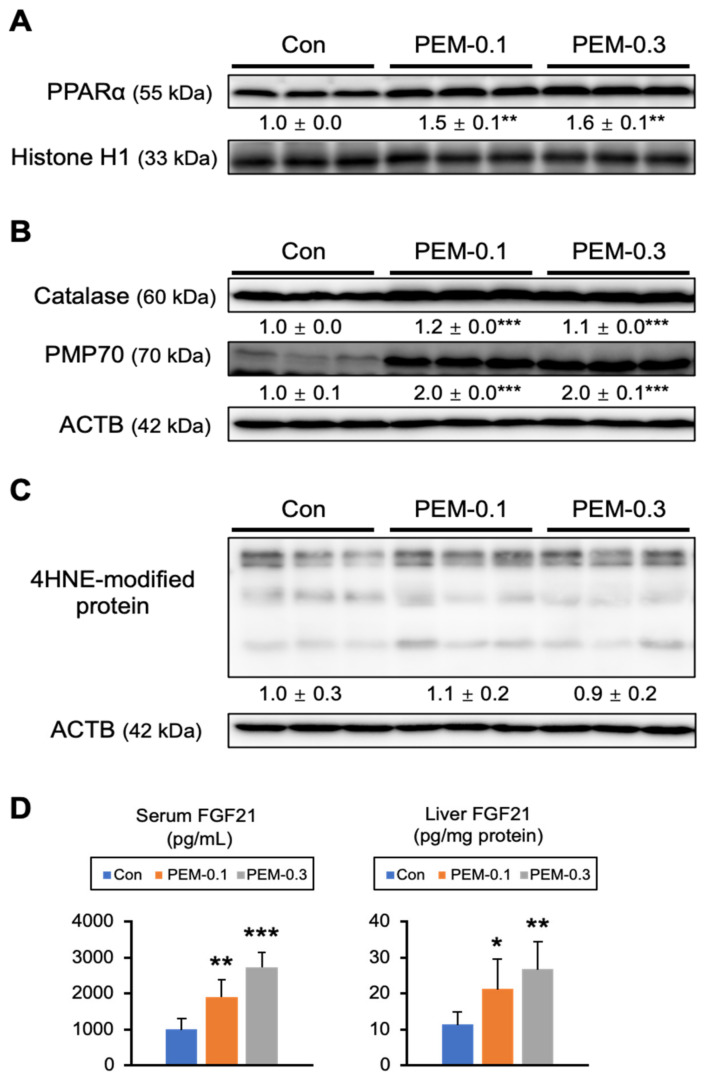
Clinically relevant dose of PEM activates hepatic PPARα without enhancing oxidative stress. (**A**) Immunoblot analysis of PPARα. Histone H1 was used as a loading control. (**B**,**C**) Immunoblot analysis of catalase, PMP70, and 4-HNE-modified protein. ACTB was used as a loading control. Band intensity was measured densitometrically, normalized to that of histone H1 or ACTB, and expressed as values relative to those of control mice. Results were obtained from two independent immunoblot experiments. (**D**) Serum FGF21 level and hepatic FGF21 content. Data are expressed as the mean ± SD. * *p* < 0.05, ** *p* < 0.01, and *** *p* < 0.001 vs. control group. Con, vehicle-treated mice; PEM-0.1, pemafibrate-treated mice at a clinically relevant dose (0.1 mg/kg/day); PEM-0.3, pemafibrate-treated mice at a relatively high dose (0.3 mg/kg/day).

**Figure 9 biomedicines-10-01667-f009:**
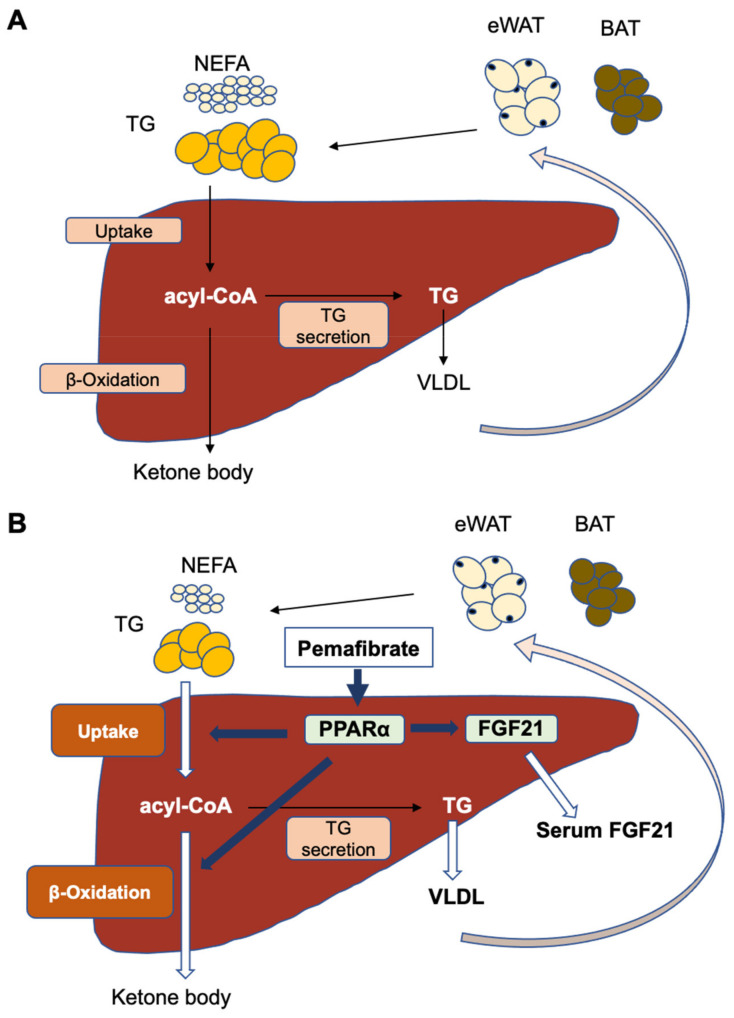
Proposed pharmacological effect of PEM in the clinical setting. (**A**) Constitutive FA/TG metabolism. (**B**) Clinically relevant dose of PEM targets hepatic PPARα to enhance the uptake of non-esterified fatty acid (NEFA) and FA β oxidation in the liver. PEM also promotes hepatic MTP expression, thus accelerating TG secretion from the liver into the systemic circulation. Additionally, PEM increases circulating FGF21 by augmenting FGF21 production in the liver.

## Data Availability

The data supporting reported results were generated during the current study.

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
