# Peer review of "Clinically Relevant Dose of Pemafibrate, a Novel Selective Peroxisome Proliferator-Activated Receptor α Modulator (SPPARMα), Lowers Serum Triglyceride Levels by Targeting Hepatic PPARα in Mice"

_biomedicines, 2022, doi:10.3390/biomedicines10071667_

Round 1

Reviewer 1 Report

The study is important considering the targeted pathology. It is original, very well conducted and the results are authentic and well documented. The practical potential for future use of these findings in human practice can be relevant.

Author Response

Reviewer 1

The study is important considering the targeted pathology. It is original, very well conducted and the results are authentic and well documented. The practical potential for future use of these findings in human practice can be relevant.

Response

Thank you very much for your suggestion. We also revised the English before submission.

Reviewer 2 Report

The authors have used doses in mice giving similar Cmax and AUC values as the usual therapeutic doses in humans. Half-lives have not been cited. However, this dose cannot be defined as “clinically relevant” in mice as no clinical response has been measured. 

Conversion of human doses to experimental animals is often calculated by the Reagan-Shaw formulae or similar equations which adjust for body weight or body surface area. This calculation would give a very different dose to that used in this study. Which calculation method is most relevant, and why?

Line 17 (and line 51): The authors refer to a superior binding efficiency of pemafibrate to fibrates – what does this mean? Pemafibrate is certainly more potent (referred to as 2500 fold) but does it have a different efficacy? Is it more selective for PPARalpha? 

Lines 68-69: pemafibrate dosed for 4 weeks in pharmacokinetic study, but only for 2 weeks in the current study (lines 92-93) – why was the time shortened in this study?

This study was in mice on a normal diet. The fibrates are used in hypercholesterolaemic patients so is the mechanism defined in this study relevant for patients with normal cholesterol and triglyceride concentrations? Would this study be more relevant if conducted in mice with high triglyceride concentrations? 

Figure 2: serum NEFA should be given in SI units of mmol/L.

Line 277: add “are” after “which”.

Author Response

Reviewer 2

The authors have used doses in mice giving similar Cmax and AUC values as the usual therapeutic doses in humans. Half-lives have not been cited. However, this dose cannot be defined as “clinically relevant” in mice as no clinical response has been measured.

Minor points

  1. Conversion of human doses to experimental animals is often calculated by the Reagan-Shaw formulae or similar equations which adjust for body weight or body surface area. This calculation would give a very different dose to that used in this study. Which calculation method is most relevant, and why?

Response

Thank you for your valuable advice. In this experiment, we did not use the conversion formula you proposed. The Cmax and AUC in clinical dose (0.2 mg twice daily for one week) are similar to those when 0.075 mg/kg and 0.1mg/kg of pemafibrate were administered to mice for four weeks, as describe in the Introduction section. Therefore, the dosage of pemafibrate was set at 0.1 mg/kg/day in this study. However, as you suggest, it is important to consider body weight and body surface area in designing doses. We added your instructions to the Discussion section as a limitation of this study.

  1. Line 17 (and line 51): The authors refer to a superior binding efficiency of pemafibrate to fibrates – what does this mean? Pemafibrate is certainly more potent (referred to as 2500 fold) but does it have a different efficacy? Is it more selective for PPARalpha?

Response

Pemafibrate has higher subtype selectivity and PPARα-binding efficiency than other fibrates, leading to a greater PPARα activation potential (very low EC50 value). So, we can say pemafibrate is more effective and selective than other PPARα activators, such as fibrates. We added the related descriptions to the Introduction section.

3. Lines 68-69: pemafibrate dosed for 4 weeks in pharmacokinetic study, but only for 2 weeks in the current study (lines 92-93) – why was the time shortened in this study?

Response

When the drug is administered for a long time, several secondary/tertiary effects occur, such as induction of drug-metabolizing enzymes and adaptation/compensation between organs. We wanted to clarify whether pemafibrate can actually reduce serum lipids also in mice in the clinically available dose and its precise mechanism (what organ is an initial and main target of pemafibrate). Therefore, we chose 2-week treatment.

  1. This study was in mice on a normal diet. The fibrates are used in hypercholesterolaemic patients so is the mechanism defined in this study relevant for patients with normal cholesterol and triglyceride concentrations? Would this study be more relevant if conducted in mice with high triglyceride concentrations?

Response

Thank you very much. We added the related descriptions and references to the Discussion section as a study limitation.

  1. Figure 2: serum NEFA should be given in SI units of mmol/L.

Response

We changed the SI units in the diagram.

  1. Line 277: add “are” after “which”.

Response

We have modified it according to your suggestions. Additionally, we have corrected grammatical errors throughout the manuscript.